# Privacy-Preserving of Deep Learning Queries by Domain Shifting

## Abstract

In the era of cloud-based deep learning (DL) services, data privacy has become a critical concern, prompting some organizations to restrict the use of online AI services. This work addresses this issue by introducing a privacy-preserving method for DL model queries through domain shifting in the input space. We develop an encoder that strategically transforms inputs into a different domain within the same space, ensuring that the original inputs remain private by presenting only the obfuscated versions to the DL model. A decoder then recovers the correct output from the model's predictions. Our method keeps the authentic input and output data secure on the local system, preventing unauthorized access by third parties who only encounter the obfuscated data. Comprehensive evaluations across various oracle models and datasets demonstrate that our approach preserves privacy with minimal impact on classification performance.

## 1 Introduction

Cloud-based deep learning services offer powerful capabilities but also introduce significant privacy risks related to user data. Uploading data for analysis or model training exposes sensitive information to third-party cloud providers, increasing the risk of data breaches, unauthorized access, or misuse of personal, proprietary, or confidential information. Additionally, service providers may retain user data and potentially re-purpose it for activities beyond the original analysis (He et al., 2022). These concerns are especially critical in fields such as healthcare, finance, and personal communications, where data protection is paramount.

There are several approaches to address the privacy issues, each with its own advantages and limitations. One common approach is to perturb the data before sharing it with model providers, using *Differential Privacy* (Dwork et al., 2006) to control the level of perturbation. Differential privacy has been effectively applied during the training process to ensure data privacy in deep learning(Wei et al., 2020). However, this approach is less suitable for protecting privacy during the inference stage, as the added noise significantly degrades model accuracy and reduces utility.

Another prominent method is *Homomorphic Encryption* (Rivest & Dertouzos, 1978; Gentry, 2009), which allows inference operations to be performed directly on encrypted data. Only the final results are decrypted by users. Despite its potential, fully homomorphic encryption (FHE) poses substantial performance challenges, with operations on encrypted data being thousands to millions of times slower than operations on plaintext (Akram et al., 2024; Jiang & Ju, 2022). This makes FHE impractical for many real-world applications. To mitigate these performance issues, multi-party computation (MPC) has been combined with homomorphic encryption to reduce DL model inference latency (Liu et al., 2017; Juvekar et al., 2018). These approaches, however, all require the model providers or third parties to implement the privacy protection mechanisms, limiting user control over the privacy of their data.

In this work, we propose a novel privacy-preserving strategy that users can implement independently, without requiring any modification to the deep learning models provided by service providers. Our approach centers on input domain shifting, where the user encodes their input by shifting it to another part of the input domain. Only the obfuscated input is sent to the model, protecting the privacy of the original data from the model service provider.

We define the *significant input domain* as the subset of the input space containing all real-world samples. The remainder of the input space, which includes inputs that are unlikely to occur naturally, constitutes the *non-significant input domain*.

In-place domain shifting maps inputs within the significant input domain to other inputs also within this domain and can be applied by encoders with only blackbox access to the deep learning (DL) model. Out-of-place domain shifting, on the other hand, encodes inputs from the significant input domain to inputs within the non-significant input domain. This approach is designed for users with whitebox access to the DL model, providing highly obfuscated inputs tailored specifically to the model. In both approaches, the DL model processes the obfuscated inputs and produces obfuscated outputs, which the user then decodes to retrieve the useful inference results.

This paper encompasses the following aspects:

- An evaluation of the feasibility of applying domain shifting as an input obfuscation method, and a demonstration of domain shifting theory on pre-trained DNN models.
- A proposal of two transform training: model-specific transform training by out-of-place domain shifting, and model-agnostic transform training by in-place domain shifting.
- A comprehensive evaluation of both obfuscation methods, including comparisons across state-of-the-art pre-trained models and datasets.

## 2 BACKGROUND

With the advancement of deep learning (DL), the associated risks can be categorized into three main types: those related to the training dataset, the trained model (including its structure and parameters), and the inputs/results of predictions (He et al., 2022). This paper focuses on privacy-preserving inference in DL models, where the goal is to protect users' input data and prediction results.

Various privacy-preserving DL inference methods have been proposed in the literature, based on concepts such as Homomorphic Encryption (HE) (Gentry, 2009) and Multi-Party Computation (MPC) (Yao, 1982). CryptoNets (Gilad-Bachrach et al., 2016) introduced the first fully HE-based neural network capable of operating on encrypted data. However, this approach incurs significant computational overhead, leading to substantial inference latency. To reduce this latency, MPC has been combined with HE to create hybrid neural network designs that preserve privacy (Liu et al., 2017; Juvekar et al., 2018).Delphi (Mishra et al., 2020) further reduces online latency by offloading most of the heavy cryptographic computations to offline processing. Additional efforts to decrease latency by combining DL techniques and MPC have been proposed in (Song et al., 2023; Nie et al., 2024).

These cryptography-based private DL inference schemes require modifications to the DL model implementation, which must be carried out by the service provider. The MPC/HE hybrid schemes also involve frequent communication between the DL model and users during the inference process. In contrast, this paper explores an alternative scenario where users do not depend on the service provider to implement privacy-preserving measures. The DL model implementation remains unmodified. Instead, users encode their data through a domain shift strategy that obfuscates private information while preserving essential features, enabling the continued use of the DL classifier service.

## 3 THREAT MODEL

In our work, we investigate a scenario where users employ deep learning services provided by a service provider who acts as an honest-but-curious adversary, meaning they collect data uploaded by the users and may use these data for other purposes. The users aim to maintain the confidentiality of their data while still being able to perform inference using the service's DNN classifier.

The users have no control over the DNN model and cannot modify it in any way. Their only interaction with the model is through querying it by sending inputs and receiving output classification results.

To protect their privacy, users avoid directly submitting their real data to the DNN. Instead, they send modified data to the DNN service and rely on the resulting classification outputs. We examine two levels of model access for users within this context:

1. **Whitebox Access**: Users have access to the internal model parameters, including detailed logit scores for each class, and can calculate gradients.

2. **Blackbox Access**: Users have no access to the internal workings of the model and receive only the output class label for the most likely class corresponding to the input data.

In the blackbox scenario, the DNN classifier takes an input $x^{in}$ and outputs to users only the most likely class label $y^{out}$ for $x^{in}$. The DNN classifier, provided by the commercial service, is assumed to be highly accurate—referred to as an oracle model. This means that when $x^{in}$ represents real data for the intended application, $y^{out}$ corresponds to the true class label $y$ with a very high probability.

In the whitebox scenario, users also receive the classification scores $s^{out}$, typically the logit values for each class corresponding to $x^{in}$. The predicted class label $y^{out}$ from the oracle model corresponds to the class with the highest score. We denote $y^{out} = f(x^{in})$ and $s^{out} = \bar{f}(x^{in})$, where and $f$ and $\bar{f}$ are functions of the oracle model that output the classification label and the classification scores, respectively. Here, $f$ is a composite of the softmax function and $\bar{f}$. In the whitebox scenario, the parameters of $\bar{f}$ are also known to the users, enabling them to calculate gradients.

# 4 THE FRAMEWORK OF PRIVATE QUERIES

In our framework, users protect their sensitive information by not directly providing their data to the DNN model. Instead, they first employ an **input encoder** $EN$ to transform their real input data $x$ into an obfuscated form $x^{ob} = EN(x)$. This **obfuscated input** $x^{ob}$ is then uploaded to the cloud service, ensuring that service providers only have access to the obfuscated data and cannot discern the original information. The obfuscated input $x^{ob}$ is processed by the service provider's DNN model (referred to as the *oracle model*), which generates an output as either $y^{ob} = \bar{f}(x^{ob})$ or $y^{ob} = f(x^{ob})$. Users then apply an **output decoder** $DE$ to convert the **obfuscated output** $y^{ob}$ back into meaningful information $DE(y^{ob})$, allowing them to utilize advanced deep learning services while maintaining control over their sensitive data throughout the process.

The **input encoder** and **output decoder** are a pair of data mapping models that handle the transformation of input and output data. These models are trained using easily accessible public datasets, which may not be as high-quality as the datasets used to train the commercial service provider's DNN model. Although this may lead to a slight performance degradation when combined with the commercial oracle model, the minor reduction in accuracy is a trade-off for the enhanced privacy and accessibility that this method provides.

When building the encoder/decoder pair, we consider two levels of applicability. For one level, we build the pair that is specifically aimed at the oracle model at hand. This encoder/decoder pair is thus model-specific. For another level, we propose a model-agnostic method, constructing encoders and decoders solely based on the dataset, independent of the oracle model.

To address these considerations, we introduce two fundamental approaches: *Model-specific Transform Training* and *Model-agnostic Transform Training* by Domain Shifting.

## 4.1 DOMAIN SHIFTING

In a Deep Neural Network (DNN) model, the input space is the set of all possible inputs that the model can accept. This could be the space of all possible pixel values for images of a specific size and color depth. The input domain, on the other hand, narrows down this space to the specific subset of inputs that are relevant and meaningful for the task at hand. It encompasses the range of values and types of data that the model is designed to process and that are expected to occur in real-world applications.

Basically, it contains two basic shifting methods:

- *Out-of-place Domain Shifting*: In this approach, inputs are transformed into an alternative domain distinct from their original domain. This resulting domain may be unrecognizable and not meaningful to human observers. This approach requires the whitebox access to the oracle model to build the *Model-specific Transform Training* upon Out-of-place Domain Shifting.

- *In-place Domain Shifting*: This method involves shifting inputs only within their original domain. The shift can occur either within the same class or across different classes. We build the *Model-agnostic Transform Training* based on In-place Domain Shifting which only requires the blackbox access.

## 4.2 MODEL-SPECIFIC TRANSFORM TRAINING WITH WHITEBOX ACCESS

For the model-specific transform training, we can conduct out-of-place domain shifting. That is, the real input can be transformed into an obfuscated input that is no longer in the original domain. Using the MNIST data set as an example, the original domain of samples from this data set is the images of handwritten digits. Figure 1 illustrates examples of images inside and outside the original domain. The upper example shows an image of handwritten digit '2' from MNIST, the original domain. Well-trained models, *Model #1* and *Model #2*, give them the same and correct predictions. The lower example shows a random generated image that does not look like any handwritten digits and is outside of the original domains. For such images outside the original domain, they do not intrinsically belong to any of the specific classes, various highly accurate classifiers (oracle models) may give them different class predictions. For example, the lower image is classified as '2' by *Model #1* and as '5' by *Model #2* in experiment reported in Section 5, as shown in the Figure 1.

When the encoder transform input $x$ to an obfuscated input $x^{ob}$ outside the original domain, the service provider of the oracle model can not discern the original input. The utility of the oracle model is kept when obfuscated inputs $x^{ob}$ from any class are designed to have their corresponding obfuscated outputs of the oracle model $y^{ob} = \bar{f}(x^{ob})$ concentrated on certain specific area. The user then use a decoder to transform the obfuscated outputs to yield class prediction $y^{DE} = DE(y^{ob})$.

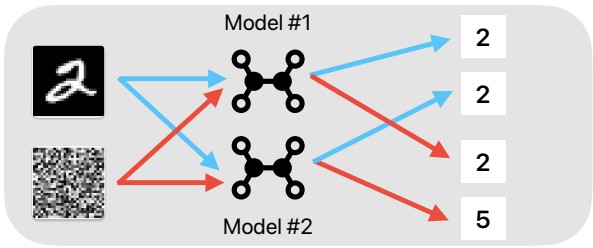

Figure 1: MNIST Out-of-place Shifting Example

For the encoder/decoder pair, we aim to keep the same functionality of the DNN classifier with or without the encoding. That is, for any input $x$, we aim to for $f(x)$ to be the same as $y^{DE} = DE \circ \bar{f} \circ EN(x)$. That is, the end-to-end classifier with privacy protection from the encoder/decoder, $DE \circ \bar{f} \circ EN$, should keep fidelity with the original oracle model. On the other hand, for privacy protection, the obfuscated input $x^{ob} = EN(x)$ should be independent of the real input $x$. Thus the training of the encoder/decoder pair can be formulated as an optimization problem with dual objectives of fidelity and obfuscation.

The model fidelity is achieved through minimization of loss

$$L_{fid} = \mathbb{E}_{x \sim p_{\text{data}}(x)} L_1[f(x), DE \circ \bar{f} \circ EN(x)],$$

where $L_1$ is taken as standard classification loss, the cross-entropy. The obfuscation is measured through the structural similarity index measure (SSIM) (Wang et al., 2004) between real input $x$ and obfuscated input $x^{ob} = EN(x)$. We choose the encoder to minimize SSIM$^2$ to achieve best obfuscation. For our implementation, we use SSIM as the loss function to train the encoder (Wang & Bovik, 2009). That is, we also want to minimize

$$L_{ob} = \mathbb{E}_{x \sim p_{\text{data}}(x)}(\text{SSIM}^2[f(x), EN(x)]) \tag{1}$$

where

$$\text{SSIM}[f(x), EN(x)] = \frac{(2\mu_{f(x)}\mu_{EN(x)} + C_1)(2\sigma_{f(x),EN(x)} + C_2)}{(\mu_{f(x)}^2 + \mu_{EN(x)}^2 + C_1)(\sigma_{f(x)}^2 + \sigma_{EN(x)}^2 + C_2)}. \tag{2}$$

In this formula, $\mu_{f(x)}$ and $\mu_{EN(x)}$ represent the mean intensity values of the original input $f(x)$ and the encoded input $EN(x)$, respectively. These terms capture the luminance similarity between the inputs. $\sigma_{f(x)}^2$ and $\sigma_{EN(x)}^2$ denote the variance of the two inputs, which reflects their contrast. $\sigma_{f(x),EN(x)}$ represents the covariance between $f(x)$ and $EN(x)$, measuring the structural similarity between the two inputs. $C_1 = (K_1 L)^2$ and $C_2 = (K_2 L)^2$ are constants used to stabilize the formula, where $L$ is the dynamic range of input values, and $K_1$ and $K_2$ are small constants, typically set to 0.01 and 0.03, respectively.

The SSIM values range from -1 to 1, where 1 indicates perfect similarity, 0 indicates no similarity, and -1 represents perfect anti-correlation. Our objective is to achieve minimal similarity between the input and the obfuscated input, thus we minimize SSIM$^2$ instead of SSIM itself.

To jointly achieve the fidelity and obfuscation, we optimize the encoder/decoder pair through minimization of the joint loss function $L_{fid} + \alpha L_{ob}$ where $\alpha$ is a tuning parameter. Through our experiments, setting $\alpha = 0.01$ generally produce good training for the encoder/decoder pair.

### 4.3 MODEL-AGNOSTIC TRANSFORM TRAINING UNDER BLACKBOX ACCESS

We now focus on developing an encoder/decoder pair that is model-agnostic, meaning it can be utilized with any oracle model (a highly accurate commercial classification model) for privacy protection.

We observe that a model-agnostic approach necessitates the encoder to perform in-place domain shifting. To understand why, consider the scenario where the obfuscated input $x^{ob} = EN(x^{in})$ falls outside the original domain. In this case, two different oracle models, $f_1$ and $f_2$, are unlikely to produce consistent classification results on $x^{ob}$ because it lacks intrinsic relevance to the classification task, as illustrated by the example in Figure 1. Consequently, we cannot rely on the same model-agnostic decoder $DE$ to accurately decode these varying results.

A model-agnostic decoder can only exist if different oracle models produce very similar obfuscated outputs $y^{ob}$ for the same obfuscated input $x^{ob}$. The key commonality across various oracle models is that they all produce the same (real) class label with high probability for inputs within the original domain. Therefore, the decoder should operate on the class label output $y^{ob} = f(x^{ob})$ rather than on the logit output $s^{ob} = \bar{f}(x^{ob})$.

We can summarize the above discussion into two necessary conditions for a model-agnostic approach: (1) the model-agnostic encoder should transform any input $x$ from the original domain into another data point $x^{ob} = EN(x^{in})$ that also remains within the original domain; and (2) the model-agnostic decoder should take as input the class label output $y^{ob} = f(x^{ob})$ from the oracle model $f$. This implies that the model-agnostic encoder/decoder pair can be constructed with blackbox access to the oracle model, as additional internal information, such as logits available through white-box access, does not provide any advantage. This is to be expected, since model internals vary among different oracle models, and thus the model-agnostic encoder/decoder should not rely on these internals.

For our implementation, we build our encoder on top of basic sub-encoders. Notice that the inputs in the original input domain is transformed to other data points within the same domain. Assume that there are total of $M$ classes in the input domain, a basic sub-encoder transforms the data among the $M$ classes according to a permutation plan. Specifically, the basic sub-encoder $EN_i$ transforms input in the $j$-th class to another data point in $j'$-th class with

$$j' = j + i \qquad mod\, M. \tag{3}$$

That is, for the basic sub-encoder $EN_0$, all inputs in the original domain are transformed to another data points within the same class. The basic sub-encoder $EN_i$ transforms inputs in the original domain to data points of another class according to a permutation plan of shifting $i$. For example, if there are 5 classes $\{0, 1, 2, 3, 4\}$, the permutation plan for $EN_1$ is $\{0 \to 1, 1 \to 2, 2 \to 3, 3 \to 4, 4 \to 0\}$, the permutation plan for $EN_2$ is $\{0 \to 2, 1 \to 3, 2 \to 4, 3 \to 0, 4 \to 1\}$, etc.

Our encoder first generates a random number $i$ from the set $0, 1, \ldots, M-1$ with equal probability, and then applies the $i$-th basic sub-encoder to produce the obfuscated output $x^{ob} = EN_i(x)$. As a result, for any input $x$ in the original domain, $x^{ob}$ randomly belongs to any class with equal probability. This ensures the privacy of $x$, as no class information about $x$ can be inferred from $x^{ob}$ alone.

The obfuscated input $x^{ob}$ is then passed to the oracle model $f$, and the resulting obfuscated output $y^{ob} = f(x^{ob})$ is returned to the user. Since the user knows the random number $i$, they can decode $y^{ob}$ to obtain the original class label using the formula:

$$y' = y^{ob} - i \qquad mod\, M. \tag{4}$$

The remaining challenge in the model-agnostic approach is constructing the $M$ basic sub-encoders: $EN_0, \ldots, EN_{M-1}$. To build these sub-encoders, we utilize generative tools such as Generative Adversarial Networks (GANs) (Goodfellow et al., 2020) and Diffusion models (Ho et al., 2020). In this paper, we focus on image classification applications as examples, so the sub-encoders will take

images as inputs and produce images as outputs, with the classes of the input and output images, $x$ and $EN_i(x)$, corresponding according to the predetermined permutation plan.

### 4.3.1 GAN MULTI-CLASS TRANSFORMATION

A Generative Adversarial Network (GAN) is a tool used to generate data within a specific domain. The generator $G$ takes a random input $z$ sampled from a probability distribution $p_z$ and produces an output $G(z)$. To ensure that $G(z)$ follows the target data distribution $p_{data}$, the generator is set up to compete with a discriminator $D$, whose goal is to distinguish real data $x \sim p_{data}$ from the generator's output. In this setup, the discriminator and the generator act as two players in a min-max game, where the objective is to solve $\min_G \max_D V(G, D)$ with the following value function:

$$V(G, D) = \mathbb{E}_{x \sim p_{data}} \log D(x) + \mathbb{E}_{z \sim p_z} \log[1 - D(G(z))]. \tag{5}$$

Both the discriminator and the generator improve through their competition, ultimately leading the generator's output $G(z)$ to closely follow the target distribution $p_{data}$.

However, for our purposes, we require a data transformer rather than a data generator. GANs can be adapted to transform data from a source distribution $p_{data}$ to a target distribution $p_{data'}$ by modifying the generator $G$ to include an additional input argument $x \sim p_{data}$ and adjusting the discriminator to distinguish between the generator's outputs and data $x' \sim p_{data'}$. In this setup, we train $G(x, z)$ in the min-max game using the following value function:

$$V(G, D) = \mathbb{E}_{x' \sim p_{data'}} \log D(x') + \mathbb{E}_{x \sim p_{data}, z \sim p_z} \log[1 - D(G(x, z))]. \tag{6}$$

Next, we consider how to implement the basic sub-encoders using GAN-based transformers. Each basic sub-encoder corresponds to a specific permutation plan: for an input $x$ in class $y$, the sub-encoder outputs $x^{ob}$ in class $per(y)$. Specifically, for $EN_i$, the permutation plan is defined as $per(y) = (y + i) \bmod M$, as given in (3).

We assume that users have access to some training data $x$ with corresponding class labels $y$ to train the basic sub-encoders. Given a class $y$, the data distribution is denoted as $p_{data|y}$.

For a single class $y_0$, we define $p_{data} = p_{data|y_0}$ as the distribution of data in class $y_0$ and set $p_{data'} = p_{data|per(y_0)}$ as the distribution of data in class $per(y_0)$. A class-$y_0$ to class-$per(y_0)$ GAN can then be trained with the objective function (6). However, this class-specific GAN does not yet constitute the desired basic sub-encoder $EN_i$, as it only transforms data from class $y_0$ to class $per(y_0)$, whereas we need it to simultaneously transform data across all classes $y$ to their corresponding classes $per(y)$.

To achieve this multi-class transformation, we condition the competing discriminator on the class label $y$. This leads to the following objective function for the min-max game:

$$V(G, D) = \mathbb{E}_{x' \sim p_{data|per(y)}} \log D(x'|per(y)) +$$
$$\mathbb{E}_{x \sim p_{data|y}, z \sim p_z} \log[1 - D(G(x, z)|per(y))]. \tag{7}$$

For training a multi-class transformation GAN, we condition on the class label $y$ using the same concept as in conditional GANs (cGANs) (Mirza & Osindero, 2014). However, our approach differs from cGANs in a key aspect: while cGANs condition the generator also on the class label, our method conditions only the discriminator on the class label. The class label is used solely during the training of the GAN to create the basic sub-encoders.

Unlike cGANs, which generate data with an explicit class label, our encoder users do not have access to the class label when encoding a data point $x$ to feed to the oracle model. Instead, they rely on the commercial oracle model to recover the class label from the encoded data.

### 4.3.2 GAN + DDPM-BASED GENERATIVE MODEL

The GAN-based encoder described above performed well on the MNIST handwritten digits dataset in the later experimental section 5. However, for more complex image datasets, the quality of the images generated by the encoder was low, which hindered accurate classification when these images were fed to oracle models. To improve the quality of the generated images, we turn to Denoising

Diffusion Probabilistic Models (DDPM) (Ho et al., 2020) , a generative tool that has recently gained popularity for producing high-quality images.

In DDPM training, the first step involves augmenting the training data from the target distribution $p_{data}$ by adding varying levels of noise. The diffusion models are then trained on this noisy data, producing a generator with parameter settings corresponding to different noise levels. For denoising, the noise level parameter is set to zero, resulting in the desired DDPM generator that produces data from the distribution $p_{data}$ without noise.

To create a multi-class transformation encoder that outputs high-quality images, we combine GANs with DDPMs. We begin by building a conditional DDPM that generates images from class $y$ when provided with the corresponding class label $y$ as input. The first layer of the conditional DDPM is an embedding layer, where each class $y$ is represented by a latent embedding vector $em_y$. We then train our multi-class transformation GAN as before, but with the goal of having the GAN output $em_{per(y)}$ instead of an image in class $per(y)$. Specifically, for an input image $x$, it first passes through the trained GAN, which outputs an embedding $em_x$. This embedding is then fed into the trained DDPM, which generates an obfuscated image $x^{ob} = EN_i(x)$. For an image $x$ in class $y$, this GAN+DDPM implementation of the basic encoder $EN_i$ produces a high-quality image $x^{ob}$ in class $per(y)$ with high probability.

It is important to note that the training of this model-agnostic encoder does not involve the oracle model.

## 5 EVALUATION

We evaluate our procedure on several common benchmark image classification datasets. MNIST dataset (Deng, 2012), Fashion-MNIST dataset (Xiao et al., 2017), CIFAR10 dataset (Krizhevsky et al., 2009), Tiny-ImageNet dataset (Le & Yang) and ImageNet dataset (Deng et al., 2009).

For each dataset, we select oracle models that serve as classifiers with accuracy comparable to state-of-the-art results reported in the literature. We then construct encoder-decoder pairs for each dataset using both model-specific and model-agnostic approaches. The privacy of the original inputs is protected through obfuscation performed by the encoder. The effectiveness of this obfuscation is assessed by measuring the Structural Similarity Index (SSIM) between the real and encoded images. The utility of the classifier under the proposed protection is evaluated by analyzing the classification results produced by the Encoder-Oracle_model-Decoder pipeline.

All experiments were conducted using PyTorch 2.3.0 on Ubuntu 18.04.6 LTS, on a machine equipped with an Intel(R) Core(TM) i5-10600K @ 4.10GHz CPU and an NVIDIA TITAN X (Pascal) GPU.

### 5.1 RESULTS OF MODEL-SPECIFIC TRANSFORM TRAINING

We begin by assessing the efficacy of Model-Specific Transform Training. The experimental setup for Model-Specific Transform Training is outlined in Table 1, where each configuration is referenced by its corresponding index. The results of these experiments are presented in Table 2.

For the MNIST dataset, we trained oracle models using three different architectures: MLP, CNN, and Vision Transformer. All three oracle models achieved over $96\%$ accuracy on the MNIST data, shown in the second column of Table 2. For the CIFAR-10 dataset, we trained a CNN as the oracle model with accuracy $88.91\%$. For the more challenging Tiny-Imagenet dataset, we fine-tune the pre-trained ImageNet classifiers (Liu et al., 2021; 2022a;b) as the oracle models. The Swin Transformer structure is Swin_V2 (TorchVision, 2024b) and the ConvNeXt structure is ConvNeXt_tiny (TorchVision, 2024a).

The encoder's primary functions are to extract extensive features from the inputs and to obfuscate these features. Its structure should be tailored to the type and complexity of the input dataset. To achieve this, we can utilize the structure of the classifier designed for the input dataset. This approach is justified because such a structure has already been proven capable of extracting sufficient features and making accurate predictions.

The decoder's role is to recover the original output from the obfuscated data, essentially performing a class-to-class permutation. Given this task, the decoder can employ a simpler structure compared

| Index | Dataset | Encoder Structure | Oracle Structure |
|---|---|---|---|
| ① | | | MLP |
| ② | MNIST | MLP | CNN |
| ③ | | | Vision Transformer (ViT_B_16) |
| ④ | CIFAR10 | CNN | CNN |
| ⑤ | | ConvNeXt | ConvNeXt |
| ⑥ | Tiny-Imagenet | Swin Transformer | Swin Transformer |
| ⑦ | | ConvNeXt | Swin Transformer |
| ⑧ | | Swin Transformer | ConvNeXt |

Table 1: Model-specific Transform Training: Experimental Configuration

| Index | Oracle Model Accuracy | Fidelity | $SSIM^2$ | Oracle Inference Time | Encoder+Decoder Inference Time |
|---|---|---|---|---|---|
| ① | 96.64% | 96.26% | $9.37 \times 10^{-5}$ | 0.156 ms | 0.368 ms |
| ② | 98.73% | 98.55% | $2.52 \times 10^{-6}$ | 0.336 ms | 0.372 ms |
| ③ | 98.55% | 98.21% | $1.94 \times 10^{-8}$ | 6.47 ms | 0.376 ms |
| ④ | 88.91% | 90.56% | $6.83 \times 10^{-6}$ | 0.582 ms | 0.687 ms |
| ⑤ | 81.18% | 86.86% | $1.03 \times 10^{-5}$ | 4.28 ms | 4.54 ms |
| ⑥ | 78.81% | 87.36% | $2.48 \times 10^{-6}$ | 13.55 ms | 12.92 ms |
| ⑦ | 78.81% | 77.22% | $4.14 \times 10^{-6}$ | 13.41 ms | 4.93 ms |
| ⑧ | 81.18% | 80.09% | $8.61 \times 10^{-7}$ | 4.63 ms | 13.03 ms |

Table 2: Model-specific Transform Training and Validation Results: The Fidelity score, $SSIM^2$ and Inference Time. The inference time is timing under the circumstance of batch size = 1.

to the encoder. In our experiments, we implemented a decoder consisting of three fully connected layers interspersed with dropout layers. The inclusion of dropout layers enhances the decoder's generalization capabilities, improving its performance on unseen data.

We used MLPs with 3 layers as the encoder and MLPs with 3 layers as the decoder for each of the oracle models on the MNIST dataset. The fidelity score, which represents the percentage of classification agreement between the Encoder-Oracle_Model-Decoder pipeline and the Oracle_Model alone, is reported in the third column of Table 2. With the encoder/decoder, the utility of the oracle models is preserved, as users can recover the classification results with high fidelity. Some encoded images from ③ are shown in Figure 2.

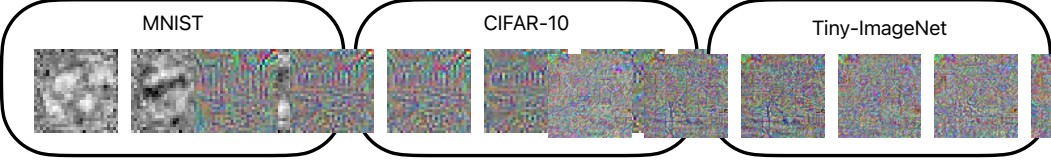

Figure 2: Out-of-place Shifting Transformation Example: Encoded Images of MNIST, CIFAR-10 and Tiny-ImageNet dataset. The encoded images are generated from ③, ④ and ⑥, respectively.

To evaluate the effectiveness of obfuscation, we calculated the structural similarity between the original image and the corresponding encoded image using SSIM (Wang et al., 2004). The SSIM measures the association between two data clusters, with values ranging from -1 to 1, where 0 indicates no association. The average of the square of SSIM ($SSIM^2$) scores over all images in the MNIST dataset is reported in the fourth column of Table 2. The very low scores indicate that the encoded images have little association with the original images, thus preserving privacy when only the encoded images are provided.

For CIFAR-10 and Tiny-Imagenet, we utilized a few multi-head attention layers or convolution layers of the oracle model to extract useful features and added a few fully connected layers as the encoder structure. The fidelity scores and the average SSIM are reported in the third and fourth columns of Table 2. The encoder/decoder pairs achieved effective obfuscation, with low SSIM scores of all dataset are close to 0, and high fidelity, with 98% of MNIST, 90% of CIFAR10 and 87% of Tiny-Imagenet for the best cases.

## 5.2 RESULTS OF MODEL-AGNOSTIC TRANSFORM TRAINING

We evaluated model-agnostic transform training on four data sets: MNIST, Fashion-MNIST, CIFAR-10 and a subset of ImageNet consisting of 20 animal classes. Notably, the training of the encoder-decoder pair does not require knowledge of the oracle models. We tested the effectiveness of the trained encoder-decoder pairs on two oracle models for each dataset. As described in Section 4.3.2, we used a GAN-based encoder for the MNIST and Fashion-MNIST datasets, while for the more complex CIFAR-10 and ImageNet datasets, the encoder was built by combining GAN and DDPM. The DDPM model in used is Stable-Diffusion-v1-4 (Podell et al., 2023). Table 3 shows the experiments' configuration and its corresponding index and their results are presented in Table 4. We developed custom classifiers as oracle models for the MNIST, Fashion-MNIST, and CIFAR-10 datasets, each with a structure tailored to its respective dataset. For ImageNet, we utilized pre-trained models from prior works Dosovitskiy et al. (2020); Liu et al. (2021). Specifically, we employed the ViT_H_14 architecture (Singh et al., 2022) from the Vision Transformer family and the Swin_V2_B structure (Liu et al., 2022a) from the Swin Transformer line.

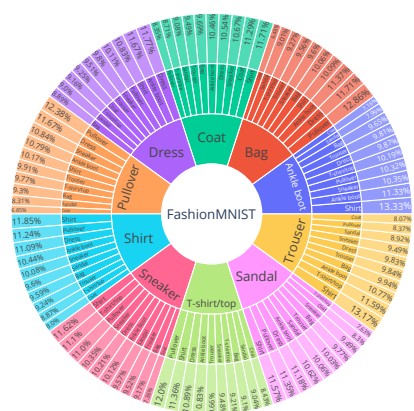

Figure 3: Pie chart of transformation in Fashion-MNIST dataset: the inner layer represents the source classes; the intermediate layer represents the target classes; the outer layer is the percentage of the transformation between source class to target class.

To evaluate the obfuscation effectiveness of the encoder, we calculated the SSIM between the true class of the input images and the oracle model's classification of the encoded images. The results are reported in the last column of Table 4. The low SSIM[2] scores indicate no association between the class of the encoded image and the original class, demonstrating a high level of obfuscation. This prevents an honest-but-curious oracle model provider from deducing the original image from the encoded image presented to it.

To further illustrate this, we present pie charts showing the distribution of encoded image classes for each original class of Fashion-MNIST dataset in Figure 3. The chart demonstrates that for each original image class, the encoded images are evenly distributed among all classes. As a result, no information about the original image class can be inferred from the encoded image.

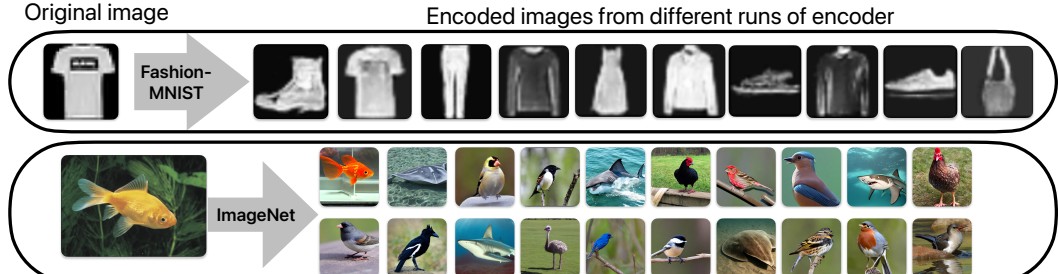

Figure 4: In-place Shifting Transformation Example: Encoded Images of Fashion-MNIST and ImageNet dataset.

Our encoder shifts the input domain, and the encoded images are generated realistic looking images from a class specified by the randomly chosen permutation plan. Figure 4 shows some examples of the encoded image. The user, knowing which permutation plan is chosen, can use oracle model classification of the encoded image to recover its classification on original image. The second and third column of Table 4 present the accuracy of between the Oracle Model alone and the Encoder-Oracle_Model-Decoder pipeline. The evaluation of Encoder-Oracle_Model-Decoder pipeline achieve almost the same accuracy as the Oracle Model on MNIST and Fashion-MNIST datasets. For more complicated dataset, such as CIFAR-10 and ImageNet, there are accuracy drops.

| Index | Dataset | Encoder Structure | Oracle Structure |
|-------|---------|-------------------|------------------|
| ❶ | MNIST | GAN | MLP |
| ❷ | | | Vision Transformer (ViT_B_16) |
| ❸ | Fashion-MNIST | GAN | CNN #1 |
| ❹ | | | CNN #2 |
| ❺ | CIFAR-10 | GAN + DDPM | CNN #1 |
| ❻ | | | CNN #2 |
| ❼ | ImageNet | GAN + DDPM | Vision Transformer (ViT_H_14) |
| ❽ | (20 classes of animals) | | Swin Transformer (Swin_V2_B) |

Table 3: Model-agnostic Transform Training: Experimental Configuration

| Index | Oracle Model Accuracy | Pipeline Accuracy | SSIM$^2$ | Oracle Inference Time | Encoder+Decoder Inference Time |
|-------|----------------------|-------------------|----------|----------------------|-------------------------------|
| ❶ | 98.12% | 97.28% | 0.0355 | 0.16 ms | 0.35 ms |
| ❷ | 98.55% | 97.49% | 0.0359 | 6.47 ms | 0.37 ms |
| ❸ | 90.97% | 90.69% | 0.0093 | 1.29 ms | 0.38 ms |
| ❹ | 89.41% | 88.12% | 0.0094 | 1.35 ms | 0.32 ms |
| ❺ | 88.91% | 80.30% | 0.0013 | 9.53 ms | 4.13 s |
| ❻ | 89.13% | 80.10% | 0.0013 | 10.7 ms | 4.12 s |
| ❼ | 88.55% | 75.10% | 0.0777 | 10.2 ms | 4.14 s |
| ❽ | 84.12% | 70.40% | 0.0768 | 12.7 ms | 4.14 s |

Table 4: Model-agnostic Transform Training and Validation Results: The Accuracy, SSIM$^2$ and Inference Time. The inference time is timing under the circumstance of batch size $= 1$ .

## 5.3 EVALUATION ON OVERHEAD

The additional overhead arises from the inference execution times of both the encoder and decoder, for both out-of-place and in-place domain shifting methods. In Table 2 and Table 4, we present two key timing measurements. The **Oracle Inference Time** indicates the duration required to process a single input query using the oracle model. The **Encoder+Decoder Inference Time** represents the additional time incurred when implementing our domain shifting method.

In the out-of-place domain shifting scenario, our protection method achieves significantly faster inference times on MNIST dataset, our method requires approximately 0.5 ms, which is substantially lower than the 0.481s reported previously in (Juvekar et al., 2018) and 3.58s in (Liu et al., 2017). Similarly, for the CIFAR-10 dataset, our method completes inference in just 1.2 ms, a marked improvement over the 472s reported in (Liu et al., 2017) and 9.74s in (Juvekar et al., 2018).

For in-place domain shifting, the GAN inference time is about 0.3 times the oracle inference time. However, the DDPM generation process takes longer, with the generation of a single image requiring around 4 seconds. Consequently, the extra time incurred after applying protection measures is approximately 4.12 seconds. Despite this, it remains much quicker than the overhead seen in previous works Liu et al. (2017); Juvekar et al. (2018). Unlike approaches that involve significant communication overhead due to layer-by-layer processing in MPC, our in-place shifting technique introduces only 4.12s of latency when querying the CIFAR-10 and ImageNet datasets, which is still faster than the 5s inference time for CIFAR-10 reported in Nie et al. (2024).

## 6 CONCLUSION

This paper presents a novel privacy-preserving approach that can be implemented by users, without requiring modifications to the deep learning (DL) models provided by service providers. The core strategy involves the development of a domain-shifting encoder by the user. Depending on their access level to the DL model, users can create either in-place shifting encoders (for model-agnostic schemes) or out-of-place shifting encoders (for model-specific schemes). These approaches enable users to perform classification tasks using DL model services while ensuring data privacy. Furthermore, the overhead on inference execution time is significantly lower compared to schemes based on homomorphic encryption (HE), which require modifications by the service provider, unlike our approach.

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
