# OpenReview forum: "Privacy-Preserving of Deep Learning Queries by Domain Shifting"
_ICLR.cc/2025/Conference — Submitted to ICLR 2025_

### Official Review · Reviewer_TjH2 · 2024-10-31

**Soundness:** 1
**Presentation:** 3
**Contribution:** 1
**Rating:** 1
**Confidence:** 5

**Summary:**

This paper proposes a framework for protecting the privacy of inputs during inference. The key insight is to transform the input into a different domain using an encoder before sending it to an external service provider to perform inference. A decoder is then used to transform the model's prediction to recover the correct output. The authors evaluate their method on several image classification datasets (MNIST< CIFAR-10, Imagenet)

**Strengths:**

1. The paper is easy to understand
2. Evaluation is done on several image classification datasets.

**Weaknesses:**

In the model-agnostic setting, the authors propose to transform the input image from its original class (class $j$) to a different class(class $j'=j+i \mod M$). Since $i$ is chosen randomly, it wouldn't be possible for the service provider to know the original class. Unfortunately, there are two major flaws with this approach:

1.  Compute: Doing this requires a lot of compute (likely on the order of compute used to perform the inference itself!). This makes it impractical in most remote-inference settings, where the user does not have the compute resources necessary to perform inference (let alone complex input transformations) locally.

2. Access to Training data: The authors assume that the user has access to labeled training data. This data is required to train the GAN (section 4.3.1) to perform the transformation of the input from one class to another requires access to labeled data. Note that this labeled data could have been used to train an inference model and perform inference locally, sidestepping the problem of performing privacy-perserving remote inference! Additionally, the  accuracy of this whole pipeline is bottlenecked by the ability of the GAN to correctly transform the input image from one label to another, so I'm not convinced that the proposed approach is better than just training a model on the user's end and performing inference locally.


In the model specific setting, the authors propose using a network to transform the input in a way that it reduces the SSIM between the original and transformed image.Reduction of SSIM does not guarantee privacy. More rigorous privacy guarantees involve adding noise to the encoded input [1]. As far as I can tell, the proposed method provides no privacy guarantees.

[1] Maeng et al. "Bounding the Invertibility of Privacy-preserving Instance Encoding using Fisher Information" NeurIPS 2023

**Questions:**

Can you please address the points raised under weaknesses?

---

> ### Author Response · Authors · 2024-11-27
>
> We greatly appreciate the time and effort you dedicated to reviewing our work. Your valuable feedback has highlighted several important points that we will address in detail below:
>
>
> 1. Computational requirements in the model-agnostic setting
>
> Thank you for raising the concern about computational overhead. We acknowledge that transforming input images through complex operations, such as training GANs, can be resource-intensive. In our experiments, we demonstrated that the training process could be completed within one week using a single GPU. While this may seem demanding, it is a one-time cost for the user, and the trained model can then be used repeatedly for inference.
> We believe this trade-off between computational effort and privacy preservation is practical for users with sufficient local computational resources. However, we acknowledge that this might not be feasible for all users, and we will address this limitation explicitly in the revised manuscript.
>
>
> 2. Access to labeled training data
>
> You raise an important point regarding the need for labeled training data. Our work assumes that users have access to publicly available labeled datasets for training the encoder and decoder. While we recognize that using private datasets would be ideal, such datasets are often inaccessible due to commercial or legal restrictions. Hence, we designed and evaluated our approach using public datasets, which are more widely available to users.
> Regarding the suggestion of local inference, we agree that if users have sufficient data and computational resources, training a local inference model may indeed sidestep the need for privacy-preserving remote inference. However, in scenarios where users lack the resources or expertise to train complex models for local inference, our approach provides an alternative by focusing on privacy preservation in remote settings. We will clarify this distinction in the revised manuscript.
>
>
> 3. Privacy guarantees in the model-specific setting
>
> Thank you for pointing out the importance of rigorous privacy guarantees and referencing noise-based mechanisms. While we focus on reducing the SSIM between the original and transformed images to demonstrate obfuscation effectiveness, we agree that SSIM alone does not constitute a formal privacy guarantee.
> We have considered model inversion attacks and input reconstruction attacks on the encoder-decoder mechanism. It is worth noting that these attacks are generally effective against shallow neural networks, while the encoder in our method uses a deep neural network, making successful reconstruction significantly more challenging. However, we acknowledge the need for stronger, more formal privacy evaluations. In response to your feedback, we will:
> 	1	Include experiments evaluating membership inference attacks to assess privacy leakage.
> 	2	Discuss the limitations of SSIM as a metric and explore additional metrics or methods to strengthen privacy guarantees.
> 	3	Add evaluations on potential attacks and their results in the appendix.
>
> We are grateful for your detailed review and constructive criticism. Your feedback has helped us identify areas for improvement, and we will incorporate these changes in the revised manuscript to strengthen the rigor and clarity of our work.

---

### Official Review · Reviewer_jv3w · 2024-11-03

**Soundness:** 3
**Presentation:** 3
**Contribution:** 2
**Rating:** 5
**Confidence:** 4

**Summary:**

This paper proposes a privacy-preserving strategy for deep learning queries through domain shifting. Using encoder-decoder models, it obfuscates inputs without modifying the deep learning model, supporting both whitebox and blackbox scenarios. The approach demonstrates effective privacy protection with minimal impact on performance across multiple datasets.

**Strengths:**

1. The key idea by leveraging the encoder and decoder is intersting and clear.
2. The method accommodates both blackbox and whitebox access scenarios. Model-agnostic in-place domain shifting works without internal model access, while model-specific out-of-place shifting enhances privacy when access to model internals is available.
3.Compared to homomorphic encryption, this approach significantly reduces computational costs, achieving rapid inference times even on complex datasets. It maintains privacy effectively without the high latency typically associated with other cryptographic methods.

**Weaknesses:**

1. If an attacker repeatedly sends disguised data and observes the output patterns of the model, the label replacement scheme may be gradually compromised through an inference attack, thereby undermining privacy protection. It would be interesting to explore this extreme scenario.
2. While leveraging the encoder and decoder has been discussed in [1, 2], it is important to clarify the differences between them.
[1] InferDPT: Privacy-Preserving Inference for Black-box Large Language Model
[2] FedAdOb: Privacy-Preserving Federated Deep Learning with Adaptive Obfuscation
3. It would be more valuable to explore these methods with more sensitive data, such as facial images or address data in NLP.

**Questions:**

1.The method requires that the pseudo-data can maintain consistent classification results across different models, but in practice, how to ensure that the accuracy of the decoder is not affected by the fact that the classification labels may vary somewhat due to the structural differences of different models?
2.Could the authors provide formal privacy metrics, such as leveraging membership inference attack, to quantitatively evaluate privacy instead of relying solely on empirical SSIM?

---

> ### Author Response · Authors · 2024-11-27
>
> We greatly appreciate the time and effort you put into reviewing our paper and providing valuable suggestions and questions. Below, we address your comments in detail:
>
>
> 1. Risk of inference attack through repeated disguised data and output observation
>
> Thank you for pointing out this potential risk. To mitigate the issue where an attacker could exploit repeated disguised data to infer output patterns and compromise the label replacement scheme, we propose introducing additional complexity to the obfuscation process. Specifically, users can randomly select one of several pre-defined permutation plans to obfuscate their data. This randomized approach significantly increases the difficulty for an attacker to identify consistent patterns, thereby enhancing privacy protection. We will discuss this scenario and our proposed countermeasure in the revised manuscript.
>
>
> 2. Clarification of the differences between encoder-decoder usage in [1] and [2]
>
> We appreciate your suggestion to include a discussion of the referenced papers.
> Our approach, while leveraging encoder-decoder mechanisms, focuses primarily on input obfuscation for inference rather than federated learning. This distinction will be clarified in the revised manuscript. Thank you for pointing this out, and we will consider the relevance of these papers in future iterations of our work.
>
>
> 3. Exploration with more sensitive data
>
> Thank you for this insightful suggestion. We agree that evaluating our method on more sensitive data, such as facial images or address data in NLP, would provide additional value. However, as most NLP-based privacy risks are associated with multi-query scenarios, we believe exploring obfuscation at a level higher than individual queries could be a promising direction. While this is beyond the current scope of our work, we are excited to explore these scenarios in future research and will mention this as a potential direction in the conclusion of our paper.
>
>
> 4. Ensuring consistent classification results across different models
>
> Thank you for raising this important question. To ensure consistent classification results across models with different structures, we address this by training the models with the same labels during the training phase. By aligning the training data with consistent labels, we ensure that models are highly likely to produce consistent outputs for the pseudo-data, regardless of architectural differences. This training strategy significantly minimizes variability in classification outcomes and maintains the decoder’s accuracy. We will include this explanation in the revised manuscript to clarify our approach.
>
>
> 5. Formal privacy metrics for quantitative evaluation
>
> Thank you for suggesting the inclusion of formal privacy metrics. While our current evaluation relies on SSIM to empirically measure the irrelevance between real and obfuscated inputs, we recognize the importance of leveraging formal metrics, such as membership inference attacks, to quantify privacy guarantees. We will conduct experiments using membership inference attacks in our revised evaluation and include these results to provide a more robust quantitative assessment of our method’s privacy-preserving capabilities.

---

### Official Review · Reviewer_SW6C · 2024-11-09

**Soundness:** 2
**Presentation:** 1
**Contribution:** 1
**Rating:** 3
**Confidence:** 4

**Summary:**

The researchers propose a privacy-preserving method for deep learning services that transforms input data into an obfuscated form before sending it to cloud models and then decodes the results locally, enabling the secure use of external AI services while protecting sensitive data.

**Strengths:**

The paper addresses the privacy issues in deep learning and proposes a privacy-preserving strategy that users can implement independently,
without requiring any modification to the deep learning models.

**Weaknesses:**

1. The introduction part mentions different approaches to privacy preservation, including differential privacy, holomorphic encryption, and multi-party computation. Then, they claim that their strategy is different and better. However, in their experiments part, they did not compare these existing approaches.
2. In the contribution part, the paper mentions three aspects. However, these claims are not clear. The first aspect is "An evaluation of the feasibility of applying domain shifting as an input obfuscation method and a demonstration of domain shifting theory on pre-trained DNN models." It seems it does not correspond to later sections.
3. The third aspect is "A comprehensive evaluation of both obfuscation methods, ..." However, the evaluation in the experiment part is not sufficient.
4. The threat model part is not clear. It is supposed to explain information on the adversary's side. Currently, the content in this part is not.
5. More experiments should be given to show the effectiveness of the method.

**Questions:**

See weaknesses.

---

> ### Author Response · Authors · 2024-11-27
>
> We greatly appreciate the time and effort you put into reviewing our paper and providing valuable suggestions and questions. Thank you for your constructive feedback. Below, we address your comments in detail:
>
>
> 1. Comparison with existing approaches in the evaluation part
>
> In our evaluation, we focus on three aspects for comparison with existing methods:
> 	•	Model/Pipeline Performance
> 	•	Privacy-Preserving Performance
> 	•	Temporal Overhead
> Model/Pipeline Performance: In our experiments, we provide a detailed evaluation of performance degradation across various datasets and model architectures. Unlike homomorphic encryption and multi-party computation, which are designed to operate on basic computing operations and communication protocols, our method directly interacts with deep neural networks (DNNs). Since these existing methods do not operate on the DNN model itself, they inherently do not introduce performance degradation. However, this difference highlights that they are not directly comparable to our approach in this context.
> Privacy-Preserving Performance: We evaluate privacy-preserving performance by calculating the Structural Similarity Index Measure (SSIM) between the real input and obfuscated input. This demonstrates the irrelevance between the two, ensuring that an adversary cannot recover the real input from the obfuscated one. In contrast, existing approaches like homomorphic encryption rely on the inherent complexity of encryption algorithms during communication and inference.
> Temporal Overhead: We briefly discussed the temporal overhead of our method compared to existing approaches in the overhead section. In the revised version, we will include a more detailed comparison in the appendix to further substantiate our claims.
>
>
> 2. Clarification of the threat model
>
> Regarding the threat model, we assume the service provider to be an honest-but-curious adversary. This means that while the service provider performs computations as intended, it may attempt to infer private information from the data it processes.
> On the adversary's side, they only have access to the obfuscated input and the obfuscated output. As demonstrated in the evaluation section, it is challenging for the adversary to recover the real input or output from these obfuscated forms. We will clarify this assumption in the threat model section and provide additional examples to strengthen its understanding.
>
>
> We hope this addresses your concerns, and we will ensure these clarifications are incorporated into the revised manuscript. Thank you again for your thoughtful comments and for helping us improve our work.

---

### Meta-Review · Area_Chair_7rYR · 2024-12-22

**Metareview:**

This paper proposes a privacy-preserving strategy for deep learning queries through domain shifting. Using encoder-decoder models, it obfuscates inputs without modifying the deep learning model, supporting both whitebox and blackbox scenarios. The approach demonstrates effective privacy protection with minimal impact on performance across multiple datasets.

Due to the addressed weaknesses of the reviewers, I vote for rejection. With incorporating these criticisms, the paper will improve significantly.

**Additional Comments On Reviewer Discussion:**

The authors rebuttal has not changed the reviewer's original assessment.

---

### Decision · Program_Chairs · 2025-01-22

Reject